



**Different physicochemical behaviors of nitrate and ammonium**
**during transport: a case study on Mt. Hua, China**
Can Wu[1], Cong Cao[2,a], Jianjun Li[2], Shaojun Lv[1], Jin Li[2,b], Xiaodi Liu[1], Si Zhang[1],
Shijie Liu[1], Fan Zhang[1], Jingjing Meng[4], Gehui Wang[1,3]*
[1] Key Lab of Geographic Information Science of the Ministry of Education, School of
Geographic Sciences, East China Normal University, Shanghai 200062, China
[2] State Key Laboratory of Loess and Quaternary Geology, Institute of Earth
Environment, Chinese Academy of Sciences, Xi'an 710061, China
[3] Institute of Eco-Chongming, Chenjia Zhen, Chongming, Shanghai 202162, China
[4] School of Environment and Planning, Liaocheng University, Liaocheng 252000,
China
[a] Now at The State University of New York at Stony Brook.
[b] Now at Institute for Environmental and Climate Research, Jinan University.
*Corresponding author. Gehui Wang (ghwang@geo.ecnu.edu.cn)



**Abstract:** To understand the chemical evolution of aerosols in the transport process,
the chemistry of $PM_{2.5}$ and nitrogen isotope compositions on the mountainside of Mt.
Hua (~1120 m a.s.l.) in inland China during the 2016 summertime were investigated
and compared with parallel observations collected at surface sampling site (~400 m
a.s.l.). $PM_{2.5}$ exhibited a high level at the surface (aver. 76.0±44.1 μg/m$^3$) and could be
transported aloft by anabatic valley winds, leading to the gradual accumulation of
daytime $PM_{2.5}$ with a noon peak at the mountainside sampling site. As the predominant
ion species, sulfate exhibited nearly identical mass concentrations in both sites, but its
$PM_{2.5}$ mass fraction was moderately enhanced by ~4% at the higher elevation. The
ammonium variations were similar to the sulfate variations, the chemical forms of both
of which mainly existed as ammonium bisulfate ($NH_4HSO_4$) and ammonium sulfate
(($NH_4$)$_2SO_4$) at the lower and higher elevations, respectively. Unlike sulfate and
ammonium, nitrate mainly existed as ammonium nitrate ($NH_4NO_3$) in fine particles and
exhibited decreasing mass concentration and proportion trends with increasing
elevation. This finding was ascribed to $NH_4NO_3$ volatilization, in which gaseous $HNO_3$
from semi-volatile $NH_4NO_3$ subsequently reacted with dust particles to form
nonvolatile salts, resulting in significant nitrate shifts from fine particles into coarse
particles. Such scavenging of fine-particle nitrate led to an enrichment in the daytime
$^{15}N$ of nitrate at the mountainside site compared with to the lower-elevation site. In
contrast to nitrate, at the higher elevation, the $^{15}N$ in ammonium depleted during the
daytime. Considering the lack of any significant change in ammonia sources during the
vertical transport process, this $^{15}N$ depletion in ammonium was mainly the result of





unidirectional reactions, indicating that additional ammonia would partition into
particulate phases and further neutralize $HSO_4^-$ to form $SO_4^{2-}$. This process would
reduce the aerosol acidity, with a higher pH (3.4±2.2) at MS site and lower ones
(2.9±2.0) at MF site. Our work provides more insight into physicochemical behaviors
of semi-volatile nitrate and ammonium, which will facilitate the improvement in model
for a better simulation of aerosol composition and properties.
**Keywords:** Ammonium; Nitrate; Stable nitrogen isotope; Haze; Volatilization

















## 1 Introduction

Atmospheric particulate matter measuring equal to or less than 2.5 μm in aerodynamic diameter ($PM_{2.5}$) is a worldwide air pollution burden that can deteriorate the urban air quality and induce adverse human health effects that contribute to lowering life expectancies (Shiraiwa et al., 2017; Lelieveld et al., 2015; Fuzzi et al., 2015; Wang et al., 2016). Recent studies have disclosed that the mechanisms underlying these effects are profoundly dependent on particle properties, e.g., the size, concentration, mixing state and chemical compositions of particles (Li et al., 2016; Liu et al., 2021; Guo et al., 2014). Thus, since 2013, China has issued strict emission directives to mitigate haze pollution. Consequently, the annual $PM_{2.5}$ concentration in China fell by approximately one-third from 2013–2017 (Zheng et al., 2018). Notwithstanding, the $PM_{2.5}$ levels in most cities in China still exceed the least-stringent target of the World Health Organization (WHO; 35 μg/m$^3$), especially in rural areas and small cities (Lv et al., 2022; Li et al., 2023).

Near-surface PM can also be transported to the upper air, and this process critically impacts radiative forcing, cloud precipitation and the regional climate by scattering/absorbing solar radiation and by influencing aerosol-could interactions (Van Donkelaar et al., 2016; Andreae and Ramanathan, 2013; Fan et al., 2018). Past assessments of these effects have been characterized by large uncertainties (Carslaw et al., 2013); for example, Bond et al. (2013) found that black carbon climate forcing varied from +0.17 W/m$^2$ to +2.1 W/m$^2$ with a 90% uncertainty. Such massive uncertainties are mainly due to our limited knowledge regarding the spatiotemporal



distribution, abundance and compositions of airborne PM (Seinfeld and Johnh, 2016;
Raes et al., 2000). In addition, aerosols may undergo aging during the vertical transport
process, causing increasingly complex compositions and changes in aerosol properties.
Despite these factors, to date, vertical observations remain comparatively scarce
compared to surface measurements. Therefore, to obtain an improved understanding of
the fundamental chemical and dynamical processes governing haze development, more
field observations of upper-layer aerosols are necessary, as these measurements could
provide updated kinetic and mechanistic parameters that could serve to improve model
simulations.
Currently, various monitoring approaches have been developed and applied to
measure vertical aerosols, e.g., satellite remote sensing and in situ lidar methods; these
approaches can be used to obtain the pollution concentration profiles (Van Donkelaar
et al., 2016; Reid et al., 2017). To accurately measure chemical compositions, aircraft
and unmanned aerial vehicles (UAVs) equipped with a variety of instruments can be
utilized in short-term sampling campaigns (Lambey and Prasad, 2021; Zhang et al.,
2017), but these tools are unsuitable for long-term continued observations due to their
high operational costs. In cases of near-surface vertical urban atmosphere observations,
techniques involving tethered balloons, meteorological towers and skyscrapers are
usually adopted (Zhou et al., 2020; Xu et al., 2018; Fan et al., 2021). However, the
vertical application range of these methods are limited to only ~500 m, thus hardly
meeting the requirements of research conducted above the boundary layer. Therefore,
high-elevation mountain sites have long been regarded as suitable places for long-term



research on the aerosol chemical compositions and properties and chemical-dynamic
processes that drive haze episodes in the lower troposphere. Although the fixed
observation position is the key drawback of this monitoring approach, it has still been
widely selected for use in various vertical observation campaigns, e.g., in past studies
conducted in Salt Lake Valley (Baasandorj et al., 2017), in Terni Valley (Ferrero et al.,
2012) and on Mt. Tai (Meng et al., 2018; Wang et al., 2011).
Mt. Hua adjoins the Guanzhong Basin of inland China, where haze pollution has
been a persistent environmental problem (Wu et al., 2020b; Wu et al., 2021; Wang et
al., 2016). In our previous studies conducted at the mountaintop of Mt. Hua, we found
that air quality was significantly affected by surface pollution, and distinctive
differences were found in the aerosol compositions and size distributions at the
mountaintop compared to those measured at lower elevations ground level (Wang et al.,
2013; Li et al., 2013). With the implementation of strict emission controls, the
atmospheric environment in this region has changed dramatically from the $SO_2$/sulfate-
dominated previous environment to the current NOx/nitrate-dominated environment
(Baasandorj et al., 2017; Wu et al., 2020c). However, the fundamental chemical and
dynamical processes driving this $PM_{2.5}$-loading explosion are unclear under the current
atmospheric state with increasing $O_3$ and $NH_3$ levels. To better rationalize these
processes, in this work, 4-hr integrated aerosol samples were synchronously collected
on the mountainside and at the lower-elevation land surface, and the chemical
components and stable nitrogen isotope compositions of nitrate and ammonium were
analyzed in the collected $PM_{2.5}$ samples. We compared the chemical compositions and



diurnal cycles between the two sampling sites and then discussed the changes in the
chemical forms of secondary inorganic ions during their vertical transport from lower
to higher elevations. Our study revealed that nitrate and ammonium exhibited distinct
physicochemical behaviors during the aerosol-aging process.
**2 Experiment**
**2.1 Sample collection**
In this campaign, the $PM_{2.5}$ samples were synchronously collected at two locations
in the Mt. Hua area during the period from 27 August to 17 September 2016. One
sampling site was located on a building belonging to the Huashan Meteorological
Bureau (34°32′N, 110°5′E) at the foot of Mt. Hua. Surrounded by several traffic arteries
and dense residential and commercial buildings, as shown in Figure 1b, this site is an
ideal urban station for studying the impacts of anthropogenic activities on local air
quality and is referred to hereafter as the "MF" site. The mountainous sampling site
(34°29′N, 110°3′E) was located approximately 8 km from the city site horizontally
(Figure 1c) at an elevation of 720 m above the average Huashan town level of ~400 m
(a.s.l.). This site was situated on a mountainside that experiences little anthropogenic
activity due to its steep terrain and is abbreviated hereafter as the "MS" site.
Furthermore, this location adjoins one of the larger valleys of Mt. Hua; therefore, the
measurements taken at this location were strongly affected by the lower-elevation air
pollutants transported upwards by the valley winds. At both measurement sites, aerosol
samples were collected at a 4-hr interval in prebaked (at 450°C for 6 hrs) quartz filters
using high-volume (1.13-$m^3$/min) air samplers (Tisch Environmental, Inc., USA). All



air samplers were installed on the roofs of buildings, approximately 15 m above the
local ground surface. Furthermore, size-resolved aerosol sampling was synchronously
conducted at two sites during summertime (10-22 August, 2020); and these samples
with nine size bins (cutoff points were 0.43, 0.65, 1.1, 2.1, 3.3, 4.7, 5.8 and 9.0 μm,
respectively) were collected using an Anderson sampler at an airflow rate of 28.3 L/min
for ~72 h. After sampling, the filter samples were stored in a freezer (at -18°C) prior to
analysis.
The hourly $PM_{2.5}$, NOx, $SO_2$ and $O_3$ mass concentrations were detected at the
mountainside sampling site using an E-BAM, a chemiluminescence analyzer (Thermo,
Model 42i, USA), a pulsed ultraviolet (UV) fluorescence analyzer (Thermo, Model 43i,
USA) and a UV photometric analyzer (Thermo, Model 49i, USA), respectively. At the
MF site, only $PM_{2.5}$ was monitored, using another E-BAM, while the data of the other
species were downloaded from the Weinan Ecological Environment Bureau
(http://sthjj.weinan.gov.cn/). Meteorological data characterizing both sampling sites
throughout the whole campaign were obtained from the Shaanxi Meteorological Bureau
website (http://sn.cma.gov.cn/).
**2.2 Chemical analysis**
Four punches (1.5-cm diameter) of each aerosol sample were extracted into 10-mL
Milli-Q pure water (18.2 MΩ) under sonication for 30 min. Subsequently, the extracts
were filtered with 0.45-μm syringe filters and detected for water-soluble ions ($Na^+$,
$NH_4^+$, $K^+$, $Mg^{2+}$, $Ca^{2+}$, $SO_4^{2-}$, $NO_2^-$, $NO_3^-$ and $Cl^-$) by using ion chromatography; the
detection limit for these nine ions was < 0.01 μg/mL. A DRI-model 2001 thermal–





optical carbon analyzer was used herein following the IMPROVE-A protocol to analyze
the organic carbon (OC) and elemental carbon (EC) in each $PM_{2.5}$ filter sample (in
$0.526\ cm^2$ punches). For more details regarding the utilized methods, readers can refer
to our previous studies (Wu et al., 2020b).

To quantify the stable nitrogen isotope compositions of nitrate ($\delta^{15}N$-$NO_3^-$) and

ammonium ($\delta^{15}N$-$NH_4^+$) in $PM_{2.5}$ samples, the filter samples were pretreated as
described for the water-soluble ion analysis. The ammonium in the extracts
(approximately half of the resulting solution) was oxidized by hypobromite ($BrO^-$) to
nitrite ($NO_2^-$), which was subsequently reduced by hydroxylamine ($NH_2OH$) in a
strongly acidic environment. The above product ($N_2O$) was then analyzed by a
commercially available purge and cryogenic trap system coupled to an isotope ratio
mass spectrometer (PT-IRMS). A bacterial method (*Pseudomonas aureofaciens*, a
denitrifying bacterium without $N_2O$ reductase activity) was used herein to convert the
sample $NO_3^-$ into $N_2O$, which was ultimately quantified through PT-IRMS. As revealed
in previous studies, the presence of $NO_2^-$ in aerosols may interfere with the denitrifier
method when measuring $\delta^{15}N$. Nonetheless, $NO_2^-$ generally composed tiny portions in
most of our samples and, on average, contributed <1.0% to $NO_3^-$+$NO_2^-$. Thus, we
believed that the proportion of $NO_2^-$ in the sample was too small to affect the resulting
$\delta^{15}N$ measurements based on the discussions reported by Wankel et al. (2010). More
details regarding the analytical artifact and quality control protocols can be found
elsewhere (Wu et al., 2021; Liu et al., 2014).
**2.3 Concentration-weighted trajectory (CWT) analysis**





CWT is a powerful tool used herein to reveal the potential spatial sources responsible
for the high PM$_{2.5}$ loadings measured on Mt. Hua; this method has been used previously
in similar studies (Wu et al., 2020c; Wu et al., 2020a). In this study, the CWT analysis
was conducted using the Igor-based tool coupled with hourly PM$_{2.5}$ concentrations and
12-hr air mass backward trajectories that were simulated by using the Hybrid-Single
Particle Lagrangian Integrated Trajectory (HYSPLIT) model (Petit et al., 2017).
**2.4 Theoretical calculations of the partial pressures of NH$_3$ and HNO$_3$ and the**
**dissociation constant of NH$_4$NO$_3$**
To obtain the product of the partial pressures of NH$_3$ and HNO$_3$, the NH$_4$NO$_3$
deliquescence relative humidity (DRH) was first calculated using equation (1) (Eq. 1).
The average DRH of NH$_4$NO$_3$ between the two sites was 65.0±2.9%, slightly lower
than the atmospheric RH (66.0±19.3%). As the works by Wexler and Seinfeld (1991)
and Tang and Munkelwitz (1993) revealed, aerosols are multicomponent mixtures, and
which the aerosol DRH is always lower than the DRH of the individual salts in the
particles. Thus, the actual DRH of the aerosols observed in this study would be lower
than the calculated DRH of NH$_4$NO$_3$. Based on these analyses, the particles would be
deliquescent most of the time, but for simplification, we always assumed that NH$_4$NO$_3$
was in an aqueous state, corresponding to the following dissociation reaction (R1):

$$\ln(DRH) = \frac{723.7}{T} + 1.6954 \qquad\qquad (Eq.\ 1)$$

$$NH_3(g) + HNO_3(g) \leftrightharpoons NH_4^+ + NO_3^- \qquad\qquad (R1)$$

According to the approach illustrated in the referenced work (Seinfeld and Johnh,
2016), the equilibrium constant of the dissociation reaction can be described as follows





(Eq. 2):

$$K_{AN}=\frac{\gamma_{NH_4NO_3}^2 m_{NH_4^+} m_{NO_3^-}}{p_{HNO_3} p_{NH_3}} \qquad (Eq.\ 2)$$

$$K_{AN}=4\times10^{17}exp\left\{64.7\left(\frac{298}{T}-1\right)+11.51\left[1+ln\left(\frac{298}{T}\right)-\frac{298}{T}\right]\right\} \qquad (Eq.\ 3)$$

$$ln(K_p)=118.7-\frac{24084}{T}-6.025ln(T) \qquad (Eq.\ 4)$$

where $K_{AN}$ (mol²/(kg² atm²)) is the equilibrium constant of R1 (this value is
temperature-dependent and can be calculated by Eq. 3), $\gamma_{NH4NO3}$ is the binary activity
coefficient for $NH_4NO_3$ ($\gamma_{NH4NO3}=\gamma_{NH4}\gamma_{NO3}$), and $m_{NH4+}$ and $m_{NO3-}$ are the molalities of
$NH_4^+$ and $NO_3^-$, respectively. To calculate $\gamma_{NH4NO3}$ and $m_{NH4+}m_{NO3-}$, the activity
coefficients of the corresponding ions and the aerosol water content were assessed using
the E-AIM (IV) model (http://www.aim.env.uea.ac.uk/aim/model4/model4a.php).
Combining equations (2) and (3), we obtained the product of the partial pressures of
$NH_3$ and $HNO_3$ ($P_{HNO3}P_{NH3}$), obtaining an average of ~15.2±26.0 ppb² at the MF site.
This value was within the range of values (1.0~37.7 ppb²) measured by the IGAC in
the summer of 2017 in Xi'an, a metropolitan city located in the Guanzhong Basin of
inland China that has suffered from serious haze pollution (Wu et al., 2020a). Thus, we
believe that $P_{HNO3}P_{NH3}$ variations can be assessed using the above method to a certain
extent. Furthermore, the dissociation constant of $NH_4NO_3$ (Kp, ppb²) can be calculated
as a function of temperature using Eq. 4, as was revealed by Mozurkewich (1993).
**3 Results and discussion**
**3.1 Overview of PM2.5 at both sites**
**3.1.1 Meteorological conditions and temporal variations in PM2.5 concentrations**
The temporal variations in the 4-hr PM2.5 mass concentrations, water-soluble ions





and meteorological factors measured at the two sampling sites are illustrated in Figure
2, and the comparisons of the above variables are summarized in Table 1. The average
temperature (T) and relative humidity (RH) at the MF site were 23.2±4.2 °C and
68.9±18.2% (Table 1), respectively, and these values were characterized by marked
diurnal variations, as shown in Figure 2a. However, relatively cold and moist weather
frequently occurred at the MS site, which exhibited less pronounced diurnal T and RH
variations, with variations approximately 8 °C and 6% lower than the mean values
derived at the MF site, respectively. Windy weather (wind speed: 3.2±2.0 m/s) also
prevailed at this sampling site with gusts above 10.0 m/s; this condition is conducive
to the dissipation of pollutants.
Overall, the $PM_{2.5}$ concentrations measured at the MF site varied from 22.8 μg/m$^3$ to
245.6 μg/m$^3$, with a mean value of 76.0±44.1 μg/m$^3$, approximately corresponding to
Grade II (75 μg/m$^3$) of the National Ambient Air Quality Standard in China. Even so,
the $PM_{2.5}$ levels at Huashan town (i.e., at the MF site) were still higher than those
measured in many typical megacities in the summertime, e.g., Xian (37 μg/m$^3$ in
2017) (Wu et al., 2020b) and Beijing (46.3 μg/m$^3$ in 2016) (Lv et al., 2019).
Noticeably, stagnant meteorological conditions with increasing RH (> 77%) and
relatively low wind speeds (< 2.0 m/s) occurred during the relatively late stage of
observation, leading to a buildup of high $PM_{2.5}$ loadings (78.7 μg/m$^3$ to 245.6 μg/m$^3$).
Such typical haze events last approximately 4 days (12 September to 16 September,
2016), indicating that aerosol pollution is still severe in rural towns despite the notable
air quality improvements recorded in most Chinese urban areas. A similar temporal
PM$_{2.5}$ pattern was seen at the MS site, where the average PM$_{2.5}$ concentration
(47.0±38.0 μg/m$^3$) was only 0.62-fold that at the MF site and was within the range of
that measured at the summit of Mt. Tai (37.9 μg/m$^3$ in 2016) (Yi et al., 2021) and on
Mt. Lushan (55.9 μg/m$^3$ in 2011) (Li et al., 2015) in summertime. As shown in Figure
2d, a multiday episode (mean PM$_{2.5}$: 106.3 μg/m$^3$) also appeared at the MS site during
the period from 12 September to 15 September, corresponding to the days on which
high surface pollution was recorded. This was indicative of the potential impacts of
surface pollution on air quality in mountainous areas.
**3.1.2 Diurnal variation in PM$_{2.5}$**
As shown in Figure 2c and 2d, regular diurnal PM$_{2.5}$ variations were seen throughout
the whole campaign, especially at the MS site. To reveal the differences in the daily
changes in PM$_{2.5}$ between the two sampling sites, the mean diurnal cycles of hourly
PM$_{2.5}$ and the boundary layer height (BLH) are depicted in Figure 3. At the low-
elevation site, the PM$_{2.5}$ concentration was moderately enhanced during the nighttime,
with a daily maximum (88.2±53.0 μg/m$^3$) observed at 6:00 local standard time (LST).
After sunrise, PM$_{2.5}$ exhibited a decreasing trend until ~15:00 LST, corresponding to
thermally driven boundary-layer growth. Conversely, the aerosol concentrations at the
higher-elevation site immediately increased as the boundary layer uplifted in the early
morning and peaked at 14:00 LST, when the MS site was located completely within the
interior of the boundary layer. Proverbially, anabatic valley winds prevail in
mountainous regions during the daytime. Thus, the aerosol-rich air at MF site may be
transported aloft by the prevailing valley breeze, leading to significantly enhanced





PM$_{2.5}$ levels at the MS site in short time periods. This finding was further verified by
the similar diurnal NO$_2$ pattern identified at the MS site, as illustrated in Figure S1. In
the forenoon period, continuous enhancement in the NO$_2$ level was observed at the MS
site, with a daily maximum of 14.4±53.0 μg/m$^3$ (at 11:00 LST); this maximum was ~7-
fold the early-morning NO$_2$ concentration. However, O$_3$ exhibited indistinctive
variations during this period, and this was indicative of less NO$_2$ being generated from
photochemical reactions. As mentioned above, there are no obvious anthropogenic
emission sources around the MS site; therefore, our observations indicate the
remarkable transport of pollutants from the lower ground surface to higher elevations
during the daytime.
Moreover, the PM$_{2.5}$ concentrations at the MS site exhibited less nighttime variation,
with a modest abatement (Figure 3b). The nocturnal BLH usually remained below the
elevation of the MS site; thus, the surface PM$_{2.5}$ may have contributed less to the aerosol
levels at the MS site at night. To identify the potential spatial sources of nocturnal PM$_{2.5}$
at the high-elevation site, a high-elevation (CWT) analysis was conducted. As
illustrated in Figure 4, the CWT values in the daylight hours were mostly concentrated
over the sampling site, consistent with our above discussions. However, relatively high
nighttime CWT loadings were distributed on Mt. Hua and in its surrounding regions,
indicating that regional transport may be a major source of PM$_{2.5}$ at the MS site at night.
Thus, the constituents of and variations in nocturnal PM$_{2.5}$ at the MS site may be mainly
the results of regional features.
**3.2 Characterization of water-soluble ions in PM$_{2.5}$**



### 3.2.1 Comparisons of water-soluble ions between the two sites

Figure 5 shows the fractional contributions of the chemical compositions to the
PM$_{2.5}$ at both sampling sites. As summarized in Table 1, the water-soluble ion level
(WSI, 24.0±15.0 μg/m$^3$) was comparable to that of organic matter (OM,
OM=1.6×OC) (Wang et al., 2016), with a fractional contribution of ~31% to PM$_{2.5}$
(Figure 5). At the higher-elevation site, the WSI exhibited lower values (19.5±16.0
μg/m$^3$), yet the proportion was moderately enhanced by ~6%. Notably, this elevated
contribution of WSIs was mostly attributed to secondary inorganic ions (sulfate,
nitrate and ammonium, (SNA)). Similar patterns in which the SNA mass fraction
increased with latitude within the mixing height have also been observed in Terni
Valley (central Italy) (Ferrero et al., 2012) and Salt Lake Valley (US) (Baasandorj et
al., 2017). Among the SNA components, sulfate was the predominant species,
exhibiting slight mass concentration differences between the two sampling sites
(10.1±6.4 μg/m$^3$ versus 9.0±7.1 μg/m$^3$). However, an ~4% enhancement in the mass
fraction of sulfate was measured at the higher elevation. Ammonium also exhibited a
similar feature, accounting for ~5%-7.5% of the PM$_{2.5}$. These sulfate and ammonium
mass concentration homogeneities across the two sites were indicative of the further
formation of these two ions during transport. Unlike sulfate and ammonium, nitrate
and its proportions showed opposite trends, decreasing with elevation; this was
consistent with most of the measured components. Above variation features of SNA
among two sites were found at most of moments in the campaign, except for 12-13
September with a higher SNA concentration at MS site (Figure 2e and 2f). On these


two days MS site remained outside the boundary layers (a.s.l., ~550 m), suggesting
less effect of the surface pollutants on the aerosol upper layers. While, the precursor
masses (~12.3 $\mu g/m^3$ for $SO_2$ and 8.4 $\mu g/m^3$ for $NO_2$) were insufficient to form so
such SNA at MS site. Thus, the higher SNA aloft on above two days may be mostly
driven by regional or long-range transport as indicated by CWT analysis (Figure S2).
Furthermore, distinct nitrate size distributions were also observed between the
different sites in the summertime of 2020. As illustrated in Figure S3, surface nitrate
was enriched in the fine mode, with a minor peak in the coarse fraction. However, the
high-elevation nitrate exhibited a bimodal pattern with two equivalent peaks in the
fine and coarse fractions and was well correlated with coarse mode calcium but
poorly correlated with ammonium ($R^2$=0.51). To our knowledge, ammonium nitrate, a
major form of fine-mode particulate nitrate, can be easily volatilized and converted
into gas-phase $NH_3$ and $HNO_3$. Thus, the gaseous $HNO_3$ volatilized from fine PM
may react with coarse-modal cations (e.g., $Ca^{2+}$, $Mg^{2+}$ and $Na^+$) to form nonvolatile
salts, leading to a significant nitrate shifts from fine particles to large particles. A
similar phenomenon was also found in our previous study conducted at the summit of
Mt. Hua (Wang et al., 2013). Nonvolatile sulfate was predominantly found in the fine
fraction at both sampling sites, which may support this concept. More evidence for
this hypothesis is presented below in section 3.3.

The diurnal cycles of the 4-hr sulfate, nitrate and ammonium are illustrated in

Figure S4. As shown in Figure S4, the total SNA concentration at the MF site
exhibited a morning peak from 8:00-12:00 LST; this variation was quite different



from that of $PM_{2.5}$. Such a difference between the total SNA and $PM_{2.5}$ at the MF site
could partially be attributed to the lower sampling resolution and enhanced formation
of SNA in the morning. The diurnal total SNA pattern identified at the MS site
coincided with the $PM_{2.5}$ pattern, exhibiting a daily maximum reaching ~25.0±18.0
$\mu g/m^3$ (from 12:00-16:00 LST), a 1.2-fold increase compared to that measured at the
MF site. Among the SNA components, morning peaks of nitrate and ammonium (from
8:00-12:00 LST) were also observed at the MF site. Through vertical transport, the
surface nitrate and ammonium can contribute to that at the MS site, leading to a
significant enhancement in nitrate and ammonium concentrations aloft with the
afternoon peaks during 12:00-16:00 LST. Even so, the maximum nitrate concentration
at the MS site (8.1±8.7 $\mu g/m^3$) was still lower than that measured at the MF site
(9.8±8.0 $\mu g/m^3$) due to the $NH_4NO_3$ volatilization under the transport process, while
ammonium exhibited the opposite trend. This finding was consistent with the above
discussion. Unlike nitrate and ammonium, similar diurnal variations in sulfate were
observed between the two sampling sites, with daily maxima observed from 12:00-
16:00 at both sites. The major sulfate formation pathway during the daytime in
summer is the photooxidation of $SO_2$ with an OH radical, and the formation rate
facilitated by this process is much lower than that of the nitrate formation process
(Seinfeld and Johnh, 2016; Rodhe et al., 1981). Thus, sulfate formation may occur
continuously during vertical transport, leading to smaller difference in the diurnal
cycle of sulfate between the two sites.
**3.2.2 Chemical forms of SNA at both sites**



As shown in Figure 5, the water-soluble ions considered herein mainly included
sulfate, nitrate and ammonium, which usually exist in the form of ammonium salts
($NH_4HSO_4$, $(NH_4)_2SO_4$, $NH_4NO_3$, and so on). In the $H_2SO_4$-$HNO_3$-$NH_3$
thermodynamic system, $H_2SO_4$ and $HNO_3$ are neutralized by ammonia under
ammonia-rich conditions and mainly exist as $(NH_4)_2SO_4$ and $NH_4NO_3$ in aerosols.
Conversely, $H_2SO_4$ is converted to $HSO_4^-$ in environments with relatively low $NH_3$
availabilities. Thus, $NH_4HSO_4$ and $NH_4NO_3$ may be the dominant aerosol components
under such environmental conditions (Rodhe et al., 1981; Seinfeld and Johnh, 2016).
To reveal the major SNA forms at the different sampling sites considered herein, the
theoretical ammonium concentration was calculated according to thermodynamic
equilibrium with the atmospheric sulfate and nitrate levels. The theoretical
ammonium levels were calculated as follows:

$$NH_{4\ theory}^+ = \left(\frac{[SO_4^{2-}]}{48} + \frac{[NO_3^-]}{62}\right) \times 18 \qquad \text{(Eq. 5)}$$

$$NH_{4\ theory}^+ = \left(\frac{[SO_4^{2-}]}{96} + \frac{[NO_3^-]}{62}\right) \times 18 \qquad \text{(Eq. 6)}$$

where $[SO_4^{2-}]$ and $[NO_3^-]$ represent atmospheric concentrations ($\mu g/m^3$). When
$(NH_4)_2SO_4$ and $NH_4NO_3$ are the dominant species, the $NH_4^+{}_{theory}$ can be calculated
using equation (5). In contrast, equation (6) suggests that $NH_4HSO_4$ and $NH_4NO_3$ are
abundantly present in the analyzed aerosols. Figure 6 compares the measured $NH_4^+$
concentrations with the theoretical $NH_4^+$ concentrations derived by the two equations
above. As illustrated in Figure 6(a), the slope of the observational $NH_4^+$ values against
the theoretical $NH_4^+$ values calculated using equation (6) was much closer to one at the
MF site than at the MS site, meaning that $NH_4HSO_4$ and $NH_4NO_3$ were the major



chemical forms of SNA at MF site. However, the opposite pattern was revealed at the
higher-elevation site; thus, the upper aerosols were characterized by abundant
$(NH_4)_2SO_4$ and $NH_4NO_3$. Such chemical compositions of aerosols at the MS site were
unexpected under the relatively ammonia-poor environment; the ammonia level at this
site was only ~10% that at the MF site (according to observational data collected during
the 2020 summertime). As can be inferred from earlier studies, the ammonia Henry's
law coefficients generally increase in value as the temperature decreases. Therefore, the
lower temperatures measured at the MS site would create a more favorable environment
for ammonia, thus shifting its partitioning toward the particulate phase. The $HSO_4^-$
transported from the MF site would thus be further neutralized to $SO_4^{2-}$ by this
additional ammonium during transport, leading to the significant difference observed
in the chemical forms of SNA between the two sites. Moreover, as the chemical
component change from $NH_4HSO_4$ to $(NH_4)_2SO_4$, the aerosol acidity moderately
decreased, showing a higher bulk $PM_{2.5}$ pH (3.4±2.2) at relatively clean upper layer
and a lower value (2.9±2.0) at heavily polluted grounds (Table 1). However, the
previous studies were generally recognized that the aerosol would become more acidic
when the air parcels were transported from the polluted to cleaner/remote regions (Liu
et al., 1996; Nault et al., 2021). Such a reduced aerosol acidity with increasing elevation
in our study was mainly due to the different physicochemical behaviors of the semi-
volatile species nitrate and ammonium, more discussions are included in the following
section.
**3.3 Physicochemical behaviors of nitrate and ammonium during transport**



According to the above discussion, a conceptual model illustrating the
physicochemical behaviors of nitrate and ammonium during vertical transport was
proposed to explain the chemical composition differences between the two sites. As
shown in Figure 7, surface air parcels containing abundant $NH_4HSO_4$ and $NH_4NO_3$
particles can be transported to the upper atmosphere by the prevailing valley winds,
and during this process, the volatile $NH_4NO_3$ is easily converted to gaseous $NH_3$ and
$HNO_3$. Subsequently, heterogeneous reactions of the gaseous $HNO_3$ with fugitive dust
occur, thus forming nonvolatile salts and resulting in the accumulation of nitrate on
the coarse-mode particles. However, as the temperature decreased, the ammonia that
volatilized from the fine particles or was derived from the surface can re-enter the
particulate phase through the gas–particle partition. Therefore, $(NH_4)_2SO_4$ would be
formed in the aerosol phase and would gradually replace $NH_4HSO_4$.
To investigate the likelihood of $NH_4NO_3$ volatilization during the transport process,
the dissociation constant of $NH_4NO_3$ (Kp) and the partial pressures of gas-phase $NH_3$
and $HNO_3$ were calculated in this study. More details regarding the calculation steps
of the above factors can be found in section 2.4. Based on the thermodynamic
principles presented by Stelson and Seinfeld (1982), when the product of the partial
pressures of $NH_3$ and $HNO_3$ ($P_{HNO3} \times P_{NH3}$) is greater than Kp, the equilibrium of the
system shifts toward the aerosol phase, thus increasing $NH_4NO_3$ formation. In
contrast, a relatively low $P_{HNO3} \times P_{NH3}/Kp$ value (<1) suggests that $NH_4NO_3$
dissociation is induced and that $NH_4NO_3$ is transferred to the gas phase. Figure 8
depicts the ratio of the product of the partial pressures of $NH_3$ and $HNO_3$ with



different ambient temperatures. As shown in Figure 8, approximately 85% of the
samples collected at both sampling sites were located within the region with
$P_{HNO3} \times P_{NH3}/K_p$ less than 1, demonstrating a common $NH_4NO_3$ dissociation
phenomenon during the observed period. For the samples with $P_{HNO3} \times P_{NH3}/K_p$ ratios
<1, the mean value of the MS-site ratios was approximately half that of the MF-site
ratios, indicating that $NH_4NO_3$ dissociation may be more likely at higher elevations
that at lower elevations. This finding was inconsistent with the aircraft observations
collected in the western U.S. by Lindaas et al. (2021), who revealed that
$P_{HNO3} \times P_{NH3}/K_p$ exhibited an increasing trend within 3 km (a.s.l.).
Moreover, the nitrogen isotope compositions of nitrate and ammonium in $PM_{2.5}$
were measured to further verify the conceptual model. As previously mentioned,
unlike daytime pollutants, nocturnal pollutants exhibited different sources between the
two sampling sites. Thus, their nitrogen isotope compositions were more complicated
and less comparable. However, for simplicity, only the daytime samples were
analyzed herein based on the hypothesis that the sources of the high-elevation
pollutants were the same as those of the pollutants collected at the MF site. As shown
in Figure 9, a discrepancy in the $\delta^{15}N$ value of nitrate ($\delta^{15}N\text{-}NO_3^-$) featuring more $^{15}N$-
enriched $NO_3^-$ was observed at the higher elevation, with a $p$ value less than 0.05.
This finding can be ascribed to the evaporation of a portion of the particulate $NH_4NO_3$
due to a dissociation shift in equilibrium; in this shift, the lighter $^{14}N$ was
preferentially incorporated into the atmosphere, leading to $^{15}N$ enrichment in the
remaining nitrate. Additionally, Freyer et al. (1993) revealed that gas-phase isotopic



exchanges between NO and $NO_2$ result in the enrichment of the heavier $^{15}N$ isotope in
the more oxidized form and may further affect $\delta^{15}N\text{-}NO_3^-$ through nitrate formation
reactions. The above isotopic exchange between $NO_2$ and NOx can be roughly
described as follows: $[\delta^{15}N(NO_2)\text{-}\delta^{15}N(NOx)]=(1\text{-} K)\times(1\text{-}f_{NO2})$, where $K$ and $f_{NO2}$ are
the temperature-dependent exchange constant and mole fraction of $NO_2$, respectively.
Based on trace gas observations, the $f_{NO2}$ values of the air aloft were very high due to
the frequently undetectable NO concentration, indicating a rather limited isotopic
exchange between $NO_2$ and NO. Therefore, the evaporation of particulate $NH_4NO_3$
have been the significant factor affecting the measurement of a higher $\delta^{15}N\text{-}NO_3^-$ at
the MS site than at the MF site in our observations. According to the above analysis,
the ammonium at higher elevation should theoretically be more and more enriched in
$\delta^{15}N$ with the continuous $NH_4NO_3$ volatilization. However, our observation of $\delta^{15}N\text{-}$
$NH_4^+$ did not correspond to above pattern, namely, ammonium at the MS site depleted
in $\delta^{15}N$ compared to that at MF site ($p<0.05$, Figure 9). Given the unchanged
ammonia sources, such seemingly unreasonable observations were mainly caused by
the gas-to-particle conversion of ammonia. In this process, the reversible phase-
equilibrium reactions between $NH_3(g)$ and $HNO_3(g)/HCl(g)$ would yield positive
enrichment in $\delta^{15}N$ of aerosol $NH_4^+$ (Walters et al., 2019); nevertheless, unidirectional
reactions involving $NH_3(g)$ and $SO_4^{2-}/HSO_4^-$ favored $^{15}N$ depletion in the particle
form as revealed by Heaton et al. (1997). Thereby, the lower $\delta^{15}N\text{-}NH_4^+$ values at MS
site were mostly driven by those irreversible reactions, rather than the reversible
equilibrium ones. This result further confirmed our conjecture that the additional



ammonia would partition into particulate phases and further neutralize the acidic
$NH_4HSO_4$, leading to an increasing pH at MS site compared to that at MF site. Taken
together, this compelling evidence verifies that fine-mode nitrate and ammonium
exhibit distinctly different physicochemical behaviors during their transport.
**4 Conclusions and atmospheric implications**
In this study, aerosol samples were collected at 4-hr intervals on the mountainside
of Mt. Hua, and the OC, EC, water-soluble ions and isotope compositions of nitrate
and ammonium were measured and compared with simultaneous observations taken
at a lower-elevation site (MF site). The particle mass at the MF site was
approximately 1.5-fold that at the higher elevation, and distinctly different diurnal
cycles were observed between the two sampling sites. Based on the BLH variation,
we revealed that near-surface $PM_{2.5}$ could be transported to the upper layers by the
mountain-valley breeze, leading to the gradual accumulation of pollutants on the
mountainside during the daytime.
Sulfate, the predominant species found among ions at both sampling sites,
exhibited nearly identical mass concentrations at the two sites but had a moderately
enhanced mass fraction at the higher elevation. Such homogeneity was also observed
in ammonium, which mainly existed as $NH_4HSO_4+NH_4NO_3$ and
$(NH_4)_2SO_4+NH_4NO_3$ at the lower- and higher-elevation sites, respectively. This
observation indicated the further formation of ammonium during the transport
process. Unlike sulfate and ammonium, nitrate at the MS site exhibited abated trends
in both its concentration and proportion, mainly due to the volatilization of $NH_4NO_3$.





With the help of nitrate and ammonium nitrogen isotopes, we proposed a conceptual
model to illustrate the different behaviors of nitrate and ammonium during vertical
transport; in this model, the semivolatile $NH_4NO_3$ in surface air parcels was easily
converted into gaseous $NH_3$ and $HNO_3$. Subsequently, heterogeneous reactions
occurred between this gaseous $HNO_3$ and fugitive dust, forming nonvolatile salts and
leading to a significant nitrate shift from fine particles into coarse particles. In
addition, the decreasing temperature was favorable for ammonia partitioning toward
the particle phase, and the addition of ammonium further neutralized $HSO_4^-$ to form
$SO_4^{2-}$. This process would reduce the aerosol acidity, with bulk $PM_{2.5}$ pH increasing
from 2.9±2.0 at MF site to 3.4±2.2 at MS site.

Over the past decade, the relative abundance of $NH_4NO_3$ has been enhanced in

most urban areas of China because strict emission directives have been promulgated
to abate the emission and environmental impacts of $SO_2$ (Xie et al., 2020; Song et al.,
2019). In this work, we observed that $NH_4NO_3$ volatilization was a ubiquitous
phenomenon for particles during transport, resulting in a shift in partwise nitrate from
the fine mode to the coarse fraction; this shift has also been reported in the offshore
areas of the UK (Yeatman et al., 2001). Thus, we think that considering only fine-
fraction nitrate may result in the conversion rate of NOx to nitrate being partly
underestimated at some times, especially in the summer. Moreover, the deposition
velocity of coarse particles is usually faster than that of fine particles; therefore, the
above process would appreciably elevate the deposition of N into the environment.
Indeed, abundant $NO_2$, $O_3$ and $NH_3$ co-occurrence is common in the East Asian



atmosphere, and under these conditions, secondary inorganic aerosols can be
effectively produced, leading to a $PM_{2.5}$ loading explosion in the urban atmosphere of
China (Wu et al., 2020c; Wang et al., 2016). Given this, harmonious reductions in
$NO_2$, $O_3$ and $NH_3$ will be urgent in further mitigation strategies to improve air quality
and alleviate other potential effects.

**Author contributions.** GW designed the experiment. CW, JiaL and CC collected the
samples. CW and CC conducted the experiments. CW and GW performed the data
interpretation and wrote the paper. All authors contributed to the paper with useful
scientific discussions or comments.

**Competing interests.** The authors declare that they have no conflict of interest.

**Acknowledgements.** This work was financially supported by the National Natural
Science Foundation of China (No. 42130704, 42007202), Shanghai Science and
Technology Innovation Action Plan (20dz1204000) and ECNU Happiness Flower
program.

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








**Table caption**
Table 1 Mass concentrations of species in the $PM_{2.5}$ samples and the meteorological
conditions at the two sampling sites.



**Figure captions**

Figure 1 (a) Location of the study sites in China, (b) topographic view of Mt. Hua
with the sampling sites marked, and (c) vertical views of the two sampling sites and
the horizontal distance between them. (The maps are produced by mapbox,
https://account.mapbox.com/, last access, 31 Dec. 2021).

Figure 2 Time series of the temperature (T), relative humidity (RH), boundary layer
height (BLH) and mass concentrations of $PM_{2.5}$ and the water-soluble ions in $PM_{2.5}$
during the observation period at the two sampling sites.

Figure 3 Diurnal variations in $PM_{2.5}$ and the boundary layer height (BLH) at the
different observation sites.

Figure 4 Concentration-weighted trajectory (CWT) analyses of $PM_{2.5}$ in both the
daytime (8:00-20:00) and nighttime (21:00-7:00) at the MS site.

Figure 5 Mass closure of $PM_{2.5}$ during the observed period (OM=1.6×OC).

Figure 6 Comparison of the calculated and observed $NH_4^+$ concentrations at the MF
and MS sampling sites.

Figure 7 Schematic of the physicochemical behaviors of nitrate and ammonium during
the transport process.

Figure 8 Temperature dependence of the ratio of the product of the partial pressures of
$NH_3$ and $HNO_3$ with the dry dissociation constant of $NH_4NO_3$.

Figure 9 Nitrate and ammonium $\delta^{15}N$ values at the two sampling sites in the daytime.











Table 1 Mass concentrations of species in the PM$_{2.5}$ samples, pH and the
meteorological conditions at the two sampling sites.

|  | Mountain foot | Mountainside |
|---|---|---|
| (i) Mass concentration in species (µg/m$^3$) | | |
| SO$_4^{2-}$ | 10.1±6.4 | 9.0±7.1 |
| NO$_3^-$ | 6.1±6.3 | 3.8±5.8 |
| NH$_4^+$ | 3.9±3.3 | 3.9±3.5 |
| Cl$^-$ | 0.4±0.5 | 0.4±0.5 |
| Na$^+$ | 0.7±0.8 | 1.7±3.1 |
| K$^+$ | 0.2±0.3 | 0.4±0.4 |
| Mg$^{2+}$ | 0.1±0.1 | 0.1±0.1 |
| Ca$^{2+}$ | 2.5±2.0 | 0.9±1.2 |
| OC | 14.0±4.7 | 5.0±2.8 |
| EC | 4.3±2.0 | 1.1±0.7 |
| PM$_{2.5}$ | 76.0±44.1 | 47.0±38.0 |
| pH$^a$ | 3.4±2.2 | 2.9±2.0 |
| (ii) Meteorological parameters | | |
| T (°C) | 23.2±4.2 | 15.0±2.5 |
| RH (%) | 68.9±18.2 | 62.8±20.0 |
| Wind speed (m/s) | 1.3±1.1 | 3.2±2.0 |
| Visibility (km) | 14.1±9.5 | 22.2±12.1 |

$^a$pH is predicted by the thermodynamic model (E-AIM (IV)

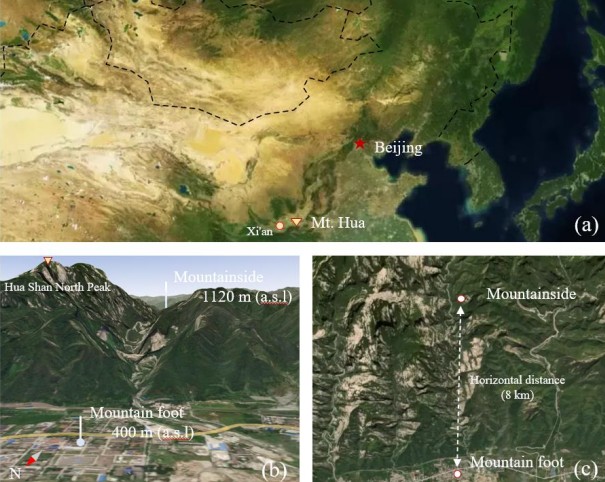


Figure 1 (a) Location of the study sites in China, (b) topographic view of Mt. Hua
with the sampling sites marked, and (c) vertical views of the two sampling sites and
the horizontal distance between them. (The maps are produced by mapbox,
https://account.mapbox.com/, last access, 31 Dec. 2021).

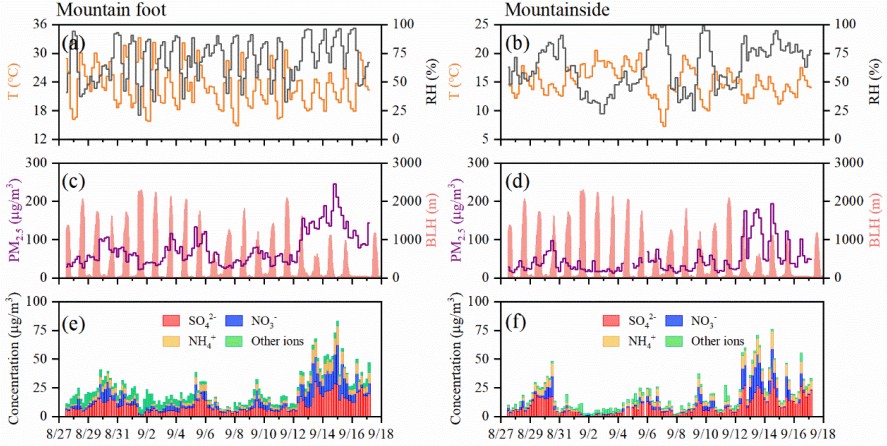


Figure 2 Time series of the temperature (T), relative humidity (RH), boundary layer
height (BLH) and mass concentrations of $PM_{2.5}$ and the water-soluble ions in $PM_{2.5}$
during the observation period at the two sampling sites.

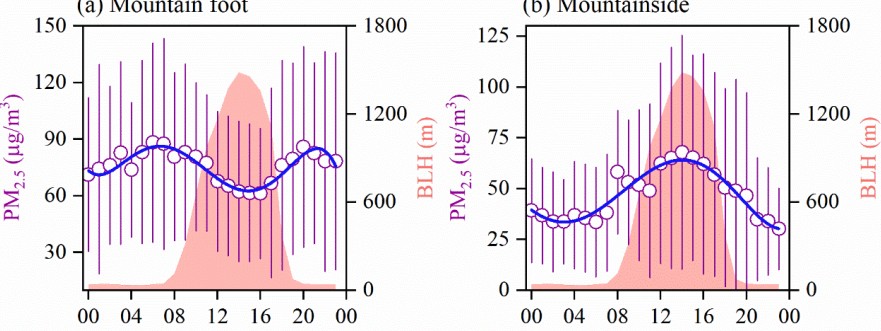

Figure 3 Diurnal variations in $PM_{2.5}$ and the boundary layer height (BLH) at the two
sampling sites.




Figure 4 Concentration-weighted trajectory (CWT) analyses of PM$_{2.5}$ in both the daytime (8:00-20:00) and nighttime (21:00-7:00) at the MS site.


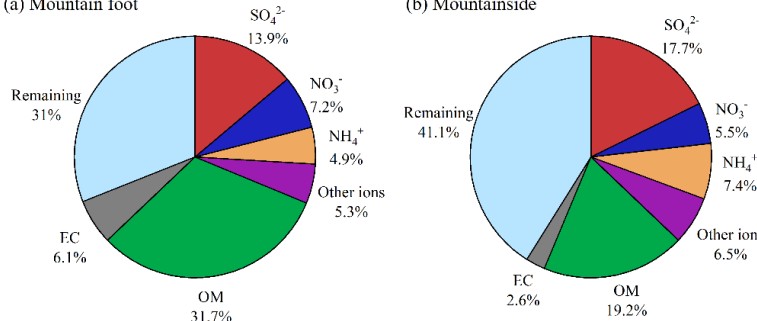


Figure 5 Mass closure of PM$_{2.5}$ during the observed period (OM=1.6×OC).







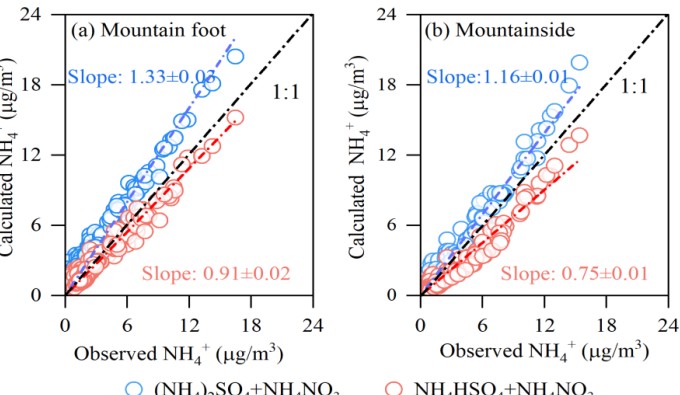

Figure 6 Comparison of the calculated and observed $NH_4^+$ concentrations at both
sampling sites.

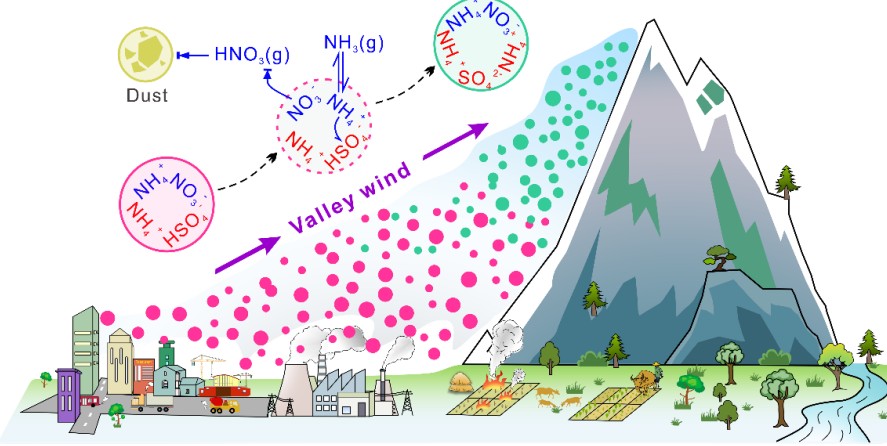

Figure 7 Schematic of the physicochemical behaviors of nitrate and ammonium during
the transport process.

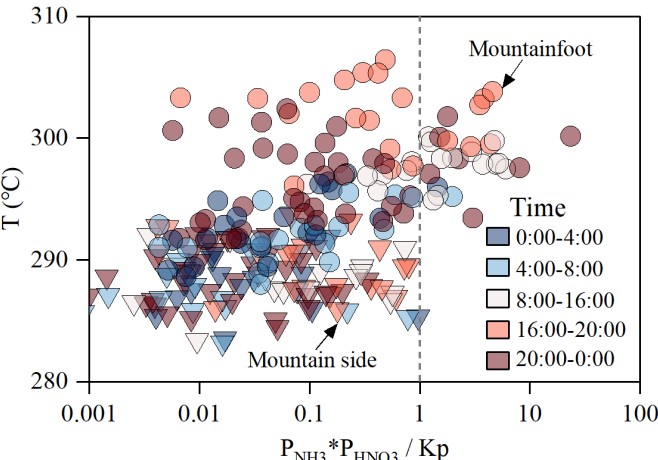

Figure 8 Temperature dependence of the ratio of the product of the partial pressures of
$NH_3$ and $HNO_3$ with the dry dissociation constant of $NH_4NO_3$.

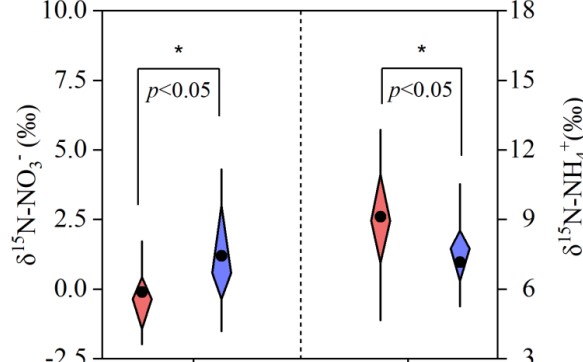

Figure 9 Nitrate and ammonium $\delta^{15}N$ values at the two sampling sites in the daytime.