# Peer review of "Different physicochemical behaviors of nitrate and ammonium"

_Atmospheric Chemistry and Physics, 2022_

## Author Comment (AC1)

Dear ACP editor:

After reading the comments from you and the reviewers, we have carefully revised our manuscript. Our responses to the comments are itemized below.

Anything for our paper, please feel free to contact Prof. Gehui Wang via ghwang@geo.ecnu.edu.cn.

All the best

Can Wu

On behalf of Prof. Gehui Wang

September 30, 2022

Reviewer(s)' Comments to Author:

**Reviewer 1**

**Comments:**

*The manuscript is a case study on the behaviour of nitrate and ammonium aerosols during transport. Based on the comparison between the chemical composition of MPs sampled simultaneously (with 4h resolution) in a mountain site and a nearby site located at low altitude, the authors describe the processes that soluble inorganic compounds undergo. During transport, the $NH_4NO_3$ volatilizes, the formed $HNO_3$ reacts with the dust, forming coarse nitrate, and the available $NH_3$ reacts with the $NH_4HSO_4$ forming $(NH_4)_2SO_4$. The isotopic study points to unidirectional reactions and additional ammonia partition into the particulate phases reducing aerosol acidity during transport. A weakness of the article is that some references of interest are missing. A very recent paper on variation of $^{15}N$ in total particulate N by Wiedenhaus et al., 2021, presents results related to the enrichment of $^{15}N$ different from those presented in this article, probably because Wiedenhaus et al have*

*determined $^{15}N$ for N total. Given the close relationship with the present manuscript, this document should be referenced and similarities/differences should be discussed.*

**Reply**: We thank the reviewer's valuable comments. We have carefully revised our manuscript according to the comments. The research conducted by Wiedenhaus et al. (2021) has been cited in the revised manuscript, and similarities was also discussed. See page 23, line 492-493, 504-507.

**Comments:**

*The volatilization of $NH_4NO_3$ and the formation of coarse nitrate is a well-known process by interaction on $HNO_3$ with coarse sodium and calcium aerosols. Authors did not mention any previous reference to these processes. Authors shall include classical references such as Harrison and Pio 1983, Pakkanen 1996, and others (line 351 and others),*

**Reply**: Suggestion taken. Those references have been cited in the revised manuscript to further support our opinion. See page 17, line 377.

**Comments:**

*The increase in $NO_3^-$ at the mountain site during September 12 and 13 is partially attributed to an external input. This may affect the interpretations of the origin of the nitrate based on the comparison between the averaged concentrations obtained in the studied sites. Therefore, this episode must be excluded for the calculation of the averages used for the interpretation of the modifications of the ions suffered during the transport from the lower levels to the mountain site.*

**Reply**: Suggestion taken. As depicted in the Figure 1, the remaindering data of SNA still exhibited the similar variation trend when excluding the episode occurred during 12-13 September, and the differences in nitrate mass concentration and fractional contribution to $PM_{2.5}$ among two sites become more pronounced. Thus, according to reviewer's advice, the data of this episode were excluded in the subsequent discussion and figures, and we also interpreted this change in the revised manuscript. See page 17, line 365-369.

[Figure]

**Figure 1** The mass concentration and fractional contributions to PM$_{2.5}$ of SNA excluding the data during 12-13 September at two sites.

**Comments:**

*Ammonia concentrations are not shown. However, some conclusions are based on the differences between the concentrations measured in MF and MS (Lines 407-408). These values, of key interest for the interpretation of data, must be presented or a reference must be included.*

**Reply**: Suggestion taken. The data of NH$_3$ has been added in the revised manuscript. See pgae 20, line 433-434.

**Minor corrections:**

*Throughout the text, the authors used the term "surface" (surface pollutants…) to refer to the low-elevation site, located in the valley. This is not appropriate. The measurements made at both sites are made at surface level, although at different heights with respect to sea level. I would distinguish between mountain and valley sites, or between high and low elevation sites.*

**Reply**: We agree with the comments above, and modified it in the revised manuscript.

**Minor corrections:**

*Line 92: WHO: add year: WHO, 2021, and add the refence in the reference list:"*

**Reply**: Suggestion taken. See page 30, line 721.

**Minor corrections:**

*Lines 155 and 159: add altitude to the coordinates: e.g. (34° 32'N. 110°5'E, 400 m a.s.l.)*

**Reply**: Suggestion taken. We have added the elevation information of the sampling site as

reviewer's advice. See page 7, line 157 and 161.

**Minor corrections:**

*Line 309: delete parenthesis*

**Reply**: Suggestion taken.

**Minor corrections:**

*Line 326: with latitude? Or with altitude?*

**Reply**: Sorry for our carelessness. Here is that the SNA mass fraction increased with altitude, and we have corrected it in the revised manuscript. See page 26, line 344.

**Minor corrections:**

*Figure S2: please, better describe the reason of including Xi'an and Dazhou in Figure S2*

**Reply**: Suggestion taken. As illustrated in Figure 2, the higher CWT loading were mainly distributed in the west and southwest areas of Mt. Hua, indicating that the high SNA aloft was may be mostly driven by the pollution long-range transport from surrounding cities. To further verify above conclusion, we chose these two cities as examples, which situates on the pollution transport pathways and suffered relatively serious haze pollution during 12-13 September. This finding was consistent with our viewpoint, and similar pattern of $PM_{2.5}$ variations also occurred in other cities on the transport pathways, e.g., Xianyang, Weinan (Figure 2). More description has been added in the revised manuscript. See page 17, line 361-365.

[Figure]

**Figure 2** Concentration-weighted trajectory (CWT) analyses of PM$_{2.5}$ during 12-13 Sep. (a). Right panel shows the time series of hourly PM$_{2.5}$ concentrations at different cities on the pollutions transport pathways (b).

**Minor corrections:**

*Line 407: add reference for observational NH$_3$ data*

**Reply**: Suggestion taken. The NH$_3$ data observed during the summertime of 2020 has been added in the revised manuscript. See page 20, line 433.

**Minor corrections:**

*Line 453-455: any explanation for these differences?*

**Reply**: The aircraft observations conducted by Lindaas et al. (2021) were mainly focused on the wildfire smoke plumes aloft, which usually contains abundant NH$_3$ and NO$x$. This would lead to a higher NH$_3$ and HNO$_3$ mixing ratio compared to that at lower elevation, and drive a higher P$_{HNO3}$×P$_{NH3}$/Kp ratio at the upper layers. Above discussions have been added in the revised manuscript. See page 22, line 486-490.

**Minor corrections:**

*Line 460-462 and figure 9: were the samples collected 12-13 September discarded? Origin of pollutant at the two sites may differs during this period.*

**Reply**: Suggestion taken. We have discarded the samples collected during 12-13 September in Figure 7, Figure 9, Figure 10 and Figure S6. As shown in the updated Figure, the recalculated results can also support our viewpoint quiet well.

**Reference**

Lindaas, J., Pollack, I. B., Calahorrano, J. J., O'Dell, K., Garofalo, L. A., Pothier, M. A., Farmer, D. K., Kreidenweis, S. M., Campos, T., Flocke, F., Weinheimer, A. J., Montzka, D. D., Tyndall, G. S., Apel, E. C., Hills, A. J., Hornbrook, R. S., Palm, B. B., Peng, Q., Thornton, J. A., Permar, W., Wielgasz, C., Hu, L., Pierce, J. R., Collett, J. L., Jr., Sullivan, A. P., and Fischer, E. V.: Empirical Insights Into the Fate of Ammonia in Western US Wildfire Smoke Plumes, J. Geophys. Res.-Atmos., 126, 10.1029/2020jd033730, 2021.

Wiedenhaus, H., Ehrnsperger, L., Klemm, O., and Strauss, H.: Stable N-15 isotopes in fine and coarse urban particulate matter, Aerosol Sci. Technol., 55, 859-870, 10.1080/02786826.2021.1905150, 2021.

---

## Author Comment (AC2)

Dear ACP editor:

After reading the comments from you and the reviewers, we have carefully revised our manuscript. Our responses to the comments are itemized below.

Anything for our paper, please feel free to contact Prof. Gehui Wang via ghwang@geo.ecnu.edu.cn.

All the best

Can Wu

On behalf of Prof. Gehui Wang

September 30, 2022

Reviewer(s)' Comments to Author:

**Reviewer 2**

**Comments:**

*The manuscript of "Different physicochemical behaviors of nitrate and ammonium during transport: a case study on Mt. Hua, China" by Wu. et al. reported the concentration of PM2.5 and composition of secondary inorganic aerosols at different altitudes. The authors investigated the physicochemical behaviors of nitrate and ammonium during the transport process. While the methods are reasonable and the conclusion convinced, there are several questions need to be clarified before publication, especially, more detail descriptions were needed.*

**Reply**: We thank the reviewer's comments. We have carefully revised our manuscript according to the comments. See details below.

**Comments:**

*Lines 39 and 65: Please change "mountain food" to "mountain foot"*

**Reply**: Sorry for our carelessness. We have corrected it and carefully checked the revised manuscript.

**Comments:**

*Line 138: The "x" in "NOx" should be italic and subscript. Please unify the expression in the text.*

**Reply**: Suggestion taken. We have unified the expression of NO$x$ as reviewer's advice in the revise manuscript. See line 140, 177, 487, 511and 569.

**Comments:**

*Line 199: "As revealed in previous studies, …". Please add references here.*

**Reply**: Suggestion taken. The reference has been added in the revised manuscript. See page 9, line 205.

**Comments:**

*Line 314: "As summarized in Table 1, the water-soluble ion level …". Please indicate which site is describing*

**Reply**: Suggestion taken. We have gave more description, see page 15, line 338.

**Comments:**

*Lin 318-320: "Notably, this elevated contribution of WSIs was mostly attributed to secondary inorganic ions (sulfate, nitrate and ammonium, (SNA))". Note that the nitrate contribution is reduced (Figure 5).*

**Reply**: Sorry for our inaccurate expression. We have changed as "Notably, this elevated contribution of WSIs was mostly attributed to sulfate and ammonium". See page 16, line 342-343.

**Comments:**

*Line 332-334: "Furthermore, distinct nitrate size distributions were also observed between the different sites in the summertime of 2020." Please add references here. And if the data is unpublished, please indicate here.*

**Reply**: The data was shown in the Figure S5, and we have indicated in the revised manuscript. See page 17, line 371.

**Comments:**

*Line 337: In addition to R2, please give p-values.*

**Reply**: Suggestion taken. See page 17, line 375.

**Comments:**

*Line 351: As Figure S3 shows, the diurnal total SNA pattern at the MS site exhibited a daily maximum value instead of MF..*

**Reply**: Sorry for our carelessness. We have corrected it in the revised manuscript.

**Comments:**

*Line 355: "..., though these peaks lagged behind those observed at the MS site by 4 hours, further substantiating the vertical transport of these pollutants." The description here is wrong.*

**Reply**: Suggestion taken. We have changed the inaccurate expression, and added more explains for peak lag at MS site. See page 18, line 394-395.

**Comments:**

*Line 393: "As can be inferred from earlier studies......" Please add references here. "Based on trace gas observations, the $f_{NO2}$ values of the air aloft were very high due to...". Please describe the data quantitatively and indicate the source of the data.*

**Reply**: Suggestion taken. We have added the reference in the revised manuscript. See page 20, line 436-437. As description in section 2.1, the data of NO$x$ at MS site was detected by a NO$x$ analyzer (Thermo, Model 42i, USA), and the NO level usually below the detection limit of the detector ($< 0.05$ ppb), leading to $>85\%$ of the samples being undetectable. Whereas, the average $NO_2$ concentration at MS site was $4.3 \pm 6.3$ μg/m$^3$, indicating a relatively high $f_{NO2}$ value

**Comments:**

*Figure 5: Remaining contributes 31-41.1% of PM$_{2.5}$, but specific species are not indicated. Figure 5(b) : Remaining is not clear in the picture.*

**Reply**: The remainder was undetected components in this study.

**Comments:**

*Figure S1 (b): The right axis title should be "O3 at MF site".*

**Reply**: Suggestion taken. We have corrected it.

---

## Author Comment (AC3)

Dear ACP editor:

After reading the comments from you and the reviewers, we have carefully revised our manuscript. Our responses to the comments are itemized below.

Anything for our paper, please feel free to contact Prof. Gehui Wang via ghwang@geo.ecnu.edu.cn.

All the best

Can Wu

On behalf of Prof. Gehui Wang

September 30, 2022

Reviewer(s)' Comments to Author:

**Reviewer 3**

**Comments:**

*This work investigated PM$_{2.5}$ and nitrogen isotope composition at a high-elevation site of Mt. Hua and a nearby surface site. By comparing the analysis results of the two sites, the authors proposed a conceptual model to illustrate the different behaviors of nitrate and ammonium during vertical transport. NH$_4$NO$_3$ decomposes into gaseous NH$_3$ and HNO$_3$ during the transport, followed by heterogeneous reactions of HNO$_3$ and dust, leading to a shift of nitrate from fine to coarse particles. This chemical process has already been well documented in many previous studies. Additionally, the partitioning of ammonia toward the particle phase during vertical transport neutralized HSO$_4^-$ from the surface and reduced the aerosol acidity. Due to the lack of direct evidence, synchronous PM characterizations at two sampling sites at different elevations can hardly explain chemical-dynamic processes. Here are my major comments.*

**Reply**: Thanks for reviewer's valuable comments on improving our work. It is generally believed in prior work that the aerosol acidity would be enhanced during the transport from the polluted to cleaner/remote regions (Nault et al., 2021; Liu et al., 1996). Whereas, our finding revealed that the aerosol acidity can also be weakened in aging process, which was mainly due to the different physicochemical behaviors of the nitrate and ammonium. Currently, numerous chemical transport models cannot well commendably the profile of nitrate and ammonium. One of the reasons is poor understanding of aerosol-related processes of above two semi-volatile salts. Our work provides more insight into physicochemical behaviors above salts. To further substantiate our finding, we have added some evidences, e.g., organic tracer and meteorological field of the sampling site. See details below.

**Comments:**

*This study assumes that the vertical transport of surface aerosols is the primary source of aerosols at the high-elevation site (MS site).*
*Besides the vertical transport, aerosols at the mountain top might also come from the subsidence of air parcel and horizontal transport, which is a possible cause for the difference in aerosol compositions and size distributions between mountaintop and ground surfaces.*
*As no atmospheric modeling was conducted to simulate the transport of air parcels, at least the meteorological field of the sampling location, we might not assure that the $NH_4NO_3$ observed at the MS area is coming from the air parcels from the MF area. Maybe the difference in PM composition is also caused by source variations, not only by the difference in physicochemical behaviors. Could the authors rule out the possibility of source variation?*

**Reply**: According to reviewer's advice, the WRF-Chem model was applied here to simulate wind filed and the divergence that represents the expansion-rate of the air mass in unit time. As shown in the vertical distribution of divergence from the near surface up to 500 hPa (Figure 1(a)), the values of divergence at MF area were greater than zero and decreased with enhanced elevation, which would drive the upward motion of the surface air parcel. And the southerly winds that would blow the pollutants into the valley prevailed at mountain foot area during the whole campaign (Figure 1(b)). These favourable meteorological conditions manifest that the surface pollutants can be transported to the upper layer by the updrafts. In addition, a westerly dominates the upper layer (above 800

hpa), of which speeds increase with enhanced elevation, indicating the significant horizontal motion of air mass in these areas. Whereas, the MS site is far below these layers, and is blocked by mountains in both east and west directions. Thus, we think that the horizontal transport may be not the major sources for the pollutants at MS site when it's inside the boundary layer.

To rule out the change of emission sources during the vertical transport, we further analyzed the organic compounds in PM$_{2.5}$ samples, e.g., levoglucosan, BkF and IP+BghiP, which are major tracers for the emissions from biomass burning, coal combustion and vehicle exhausts, respectively (Wu et al., 2020; Wang et al., 2009; Wang et al., 2007). From Figure 2, indistinctive divergences of diagnostic ratios and proportion of these organic tracers were found among both sampling sites, suggesting an insignificant change of the corresponding emission sources. Given all this, the new evidences can strongly endorse our conclusions. Above discussions have been added in the manuscript. See page 15, line 320-333.

[Figure]

**Figure 1** The distribution of averaged diurnal divergence over the whole campaign, with corresponding wind filed. (a) Longitude-pressure cross-sections at 34°29′N. (b) Horizontal distribution at surface. Wind speeds were represented by arrows sizes, and the W component of wind vectors was magnified 10 times.

[Figure]

**Figure 2** The mass ratio and proportion of organic tracers at two sampling sites.

**Comments:**

*Have the authors considered the time scales of vertical transport and atmospheric reactions mentioned in this work? Maybe the transport time between the two sites is shorter than the chemical-dynamic processes proposed in this work.*

**Reply**: As depicted in Figure 1(a), the vertical wind speed was really low with averaged value < ~0.12 m/s at the layer blow 850 hpa, indicating that the vertical motion of air mass was not as fast as expected. We simply evaluated transport time by using above speed, and found that it would take about 1.7 hours for the air parcel to move up MS site from MF site. However, as revealed by the laboratory smog chamber simulations (Liu and Abbatt, 2021; Zhang et al., 2022), the secondary ions can be rapidly generated on the initial seeds when the corresponding precursor gases were introduced in the chamber for a few minutes (Figure 3). The semblable phenomenon was also discovered in the filed observation (Wang et al., 2016). Thus, we think the chemical-dynamic processes proposed could be complete within the transport time between two sampling sites.

[Figure]

**Figure 3** Smog chamber simulation experiments for secondary ions (a and b) (Liu and Abbatt, 2021; Zhang et al., 2022). The variation of sulfate concentration during pollution episodes in Xi'an and Beijing, China (c) (Wang et al., 2016).

**Comments:**

*Lines 124-125. Are there any studies investigating chemical-dynamic processes that drive haze episodes in the lower troposphere using observations at high-elevation mountain sites? Please provide examples.*
*Such observations might only reflect the difference in aerosol chemical composition and properties between ground and high-elevation sites, but not the chemical-dynamic process.*

**Reply**: Sorry for our inaccurate expression. It was indeed scarce of the study to investigate above chemical-dynamic processes by using field observation only. To avoid ambiguity, we rewrote the sentences. See page 6, line 125-127.

**Comments:**

*Lines 330-334. Why does the similarity in mass concentrations of sulfate and ammonium at the two sites indicate further formation during the transport? Couldn't these two ions be formed at the MS site?*

**Reply**: Based on the statistical analysis, the averaged concentration of sulfate was $9.2\pm7.1$ $\mu g/m^3$. Whereas, more than 75% data had the $SO_2$ concentration below 8.0 $\mu g/m^3$ (Figure 4). Similar phenomenon was also found in the ammonia. Thus, we think that the precursor masses were insufficient to form so much sulfate and ammonium only at MS site.

[Figure]

**Figure 4** The statistical analysis of $SO_2$ concentrations measured at MS site over 8:00-19:00.

**Comments:**

*Lines 335-336, many components show increased concentrations with elevation, like $Na^+$ and $K^+$. Except for $NO_3^-$, only $Ca^+$, OC, and PM showed decreasing trends.*

**Reply**: Thank you for reminding. Due to the different filter manufacturers, the filters for

the mountainside samples collected during 8/27-8/29 have a higher background level of Na$^+$ than that of the remaining ones (Figure 5). But we inadvertently used a lower background value (i.e., the Na$^+$ concentration in blank filter for the sample collected during 8/29-9/17) to deduct the background interference for the samples in above three days. This led an overestimation of the sodium concentration, and a higher sodium concentration at MS site. In the revised manuscript, the Na$^+$ concentrations have been modified by using the correct background values, which exhibited a higher value at MF site after correction. As depicted in Figure 5, worth noting that background levels of other ions in the filters using over these three days were extremely low, and there's almost no change in their concentration especially for SNA. Furthermore, in the previous version, the rounding of the data results in the same values in Cl$^-$ and Mg$^{2+}$ concentrations at two sites.

[Figure]

**Figure 5** The chromatograms of ions extracted from different blank filters.

**Comments:**

*Lines 351-357, the shift of fine mode nitrate toward the coarse mode was well documented in previous work. Did the temperature increase with elevation? Since the dissociation of NH$_4$NO$_3$ tends to happen in warmer periods or areas, why high-elevation nitrate exhibited a bimodal pattern, but not the surface nitrate? Are the size distribution data of PM at the two sites available? The size distribution of nitrate might be closely related to the PM size distribution.*

**Reply**: The temperatures recorded at both sampling exhibited a decreasing trend with elevation. As shown in Figure S6(a), the surface nitrate was also distributed in the coarse mode that only accounts for very small fraction of the total nitrate. This feature conformed

to other surface filed observations in the summertime (e.g., Beijing, Yangtze River Delta) (Wang et al., 2015; Yang et al., 2017). At MF site with high temperature, the $NH_4NO_3$ did have a relatively high volatilization rate, even so it still needs sufficient time to achieve the volatilization amount that may change the particle size distribution. Whereas, $NH_4NO_3$ particles aloft were derived from aerosol at ground, which would be undergo volatilization and heterogeneous reaction (e.g., $HNO_3(g)$ + fugitive dust) during whole transport process. This finally results in the accumulation of nitrate on the coarse-mode particles at high elevation. Furthermore, such a bimodal mode of nitrate was also found on mountaintop of Mt. Hua as revealed in our previous study (Wang et al., 2011). Thereby, we think that above size distribution may be common at high-elevation of Mt. Hua.

**Comments:**

*Lines 367-373, could the decrease in nitrate concentration at the MS site be partly caused by dilution during the transport? As shown in Figure S1, $NO_2$ concentrations at the MS site are way lower than the MF site, and no information on $NH_3$ and $SO_2$ concentrations was available.*

**Reply**: The boundary-layer growth at daytime did lead to the nitrate concentration decrease to some extent. To eliminate the dilution effect caused by the enhanced by boundary-layer, we further discussed the proportion of nitrate in $PM_{2.5}$, which also exhibited a decreasing trend with enhanced elevation as depicted in Figure 5. Thus, we think that the decreasing nitrate mass concentration was not mainly driven by the boundary-layer growth at daytime.

**Comments:**

*Lines 405-408. Why was the MS site an ammonia-poor environment? $NH_3$ was not measured in this study, and sulfate and nitrate were almost fully neutralized by $NH_3$.*

**Reply**: Sorry for our inappropriate expression. We just want to express that the ammonia level at MS site was relatively low compared to that at MF site, according the observational data collected during 2020 summertime. The sentence has been rewritten in the revised manuscript, see page 20, line 432-437.

**Comments:**

*Lines 409-423. The authors provided a possible explanation for the change in the chemical forms of sulfate and bulk $PM_{2.5}$ pH from MF to the MS site. However, have the authors considered the change in aerosol water content from the MF to the MS site? Would the*

*changes in chemical forms of sulfate and pH be caused by variations in aerosol water content?*

**Reply**: Based on the statistical analysis, the aerosol liquid water contents (ALWC) at MS site and MF site were 26.9±71.4 μg/m³ and 27.6±63.9 μg/m³, respectively, indicating an indistinctive change among two sites (Table 1). Furthermore, we artificially enhanced the ALWC by 50%, and recalculated the aerosol pH at both sampling sites. As shown in the Figure 6, ~7% of pH was changed when the ALWC increased by 50%, which indicated that the aerosol pH in this study was insensitive to the ALWC change. Thus, we think that the change in chemical composition was the major reason for the different acidity among two sampling sites. Above discussions have been added in the revised manuscript. See page 21, line 448-453.

[Figure]

**Figure 6** The comparison of pH by changing the ALWC at both sampling sites.

**Comments:**

*Lines 425-436. The authors mentioned that the temperature decreased from MF to the MS site (line 433), why is volatile NH₄NO₃ easily converted to gaseous NH₃ and HNO₃ during the transport? Theoretically, more NH₄NO₃ should be formed through gaseous reactions at the MS site with lower temperatures.*

**Reply**: As revealed by previous studies (Arthur et al., 1982; Bergin et al., 1997), dissociation constant of NH₄NO₃ (Kp) is related not only to temperature but also to relative humidity. As the RH decrease, the Kp enhances, indicating a higher evaporation rate of NH₄NO₃ from the particles. However, the RH at MS site was lower than that at MF site, which would promote NH₄NO₃ volatilization at MS site. As depicted in the next question, the lower value of $P_{HNO3} \times P_{NH3}/Kp$ at MS (Figure 7) also conforms to above conclusion,

implying that $NH_4NO_3$ at MS site tend to remain in gaseous phase.

**Comments:**

*Lines 453 – 455, would the authors explain the inconsistent calculation compared to Lindaas et al. (2021)? Could it be attributed to the large uncertainties in the empirical calculations of $P_{HNO3}\times P_{NH3}$ and Kp?*

**Reply**: Thank you for reminding. There are indeed some uncertainties in the empirical calculations of $P_{HNO3}\times P_{NH3}$ and Kp as indicated by a wide range of $P_{HNO3}\times P_{NH3}$/Kp ratios. To minimize the uncertainty caused by data dispersion, we only choose the $15^{th} \sim 85^{th}$ percentiles of the data points ($P_{HNO3}\times P_{NH3}$/Kp<1) to further verify the result, which is same as that described in the manuscript (Figure 7). Therefore, we think that the phenomenon about a lower $P_{HNO3}\times P_{NH3}$/Kp ratio at high-elevation did exist during the campaign. The aircraft observations conducted by Lindaas et al. (2021) were mainly focused on the wildfire smoke plumes aloft, which usually contains abundant $NH_3$ and $NOx$. This would lead to a higher $NH_3$ and $HNO_3$ mixing ratio compared to that at lower elevation, and drive a higher $P_{HNO3}\times P_{NH3}$/Kp ratio at the upper layers.

[Figure]

**Figure 7** Comparison of $P_{HNO3}\times P_{NH3}$/Kp ratio among both sampling site. ($15^{th} \sim 85^{th}$ percentiles of the data points with $P_{HNO3}\times P_{NH3}$/Kp<1)

**Comments:**

*In this work, the authors always assumed that $NH_4NO_3$ was in an aqueous state. But Kp is the dissociation constant of dry salts. Will aerosol water content impact the dissociation of $NH_4NO_3$?*

**Reply**: As previously mentioned, the aerosol liquid water contents (ALWC) show an indistinctive change between MS site ($26.9\pm71.4$ $\mu g/m^3$) and MF site ($27.6\pm63.9$ $\mu g/m^3$) (*t*-

test, p=0.83). This a little discrepancy of ALWC among both sites indicated that the ALWC may be not the major driving factor for $NH_4NO_3$ dissociation or evaporation. Furthermore, the experimental discovery by Harrison et al. (1990) also demonstrated a little difference in the evaporation rates of dry $NH_4NO_3$ aerosol (-0.45Å/s) and aqueous ones (-0.49 Å/s) at 20 °C. Thus, we think that the ALWC may have a slight effect on the dissociation of $NH_4NO_3$ in our study.

.

**Comments:**

*Lines 460-494. The sources of $NH_4^+$ and $NO_3^-$ at the high-elevation site are probably not the same as those collected at the MF site, and this might lead to the difference in $\delta^{15}N$-$NH_4^+$ and $\delta^{15}N$-$NH_4^+$ values at the two sampling sites.*

**Reply**: As we answered to the first question of the reviewer, the mass ratio and proportion of organic tracers exhibited an indistinctive divergence among both sampling site, indicating an insignificant change in pollutant sources. Thus, we think that the difference in $\delta^{15}N$-$NH_4^+$ and $\delta^{15}N$-$NO_3^-$ values at the two sampling sites was mainly driven by physicochemical processes rather than the change in emission sources.

**Minor corrections**
*Lines 60-61, abstract. Define MS and MF where they first appear.*
**Reply**: Suggestion taken.

**Minor corrections**
*Lines 165-167, what type of aerosols was sampled? $PM_{2.5}$, $PM_{10}$, or TSP.*

**Reply**: Sorry for our inaccurate description. We have changed the expression as "the $PM_{2.5}$ aerosol samples with a 4-hr interval" in the revised manuscript. See page 8, line 168.

**Minor corrections**

*Line 342, typo, "so much", not "so such".*

**Reply**: Suggestion taken.

**Minor corrections**

*Line 453, typo, Change "that" to "than".*

**Reply**: Suggestion taken.

**Reference**

Arthur, W., Stelson, and, John, H., and Seinfeld: Relative humidity and pH dependence of the vapor pressure of ammonium nitrate-nitric acid solutions at 25° C, Atmos. Environ., 16, 993-1000, 1982.

Bergin, M. H., Ogren, J. A., Schwartz, S. E., and McInnes, L. M.: Evaporation of ammonium nitrate aerosol in a heated nephelometer: Implications for field measurements, Environ. Sci. Technol., 31, 2878-2883, 10.1021/es970089h, 1997.

Harrison, R. M., Sturges, W. T., Kitto, A., and Li, Y.: Kinetics of evaporation of ammonium chloride and ammonium nitrate aerosols, Atmospheric Environment Part A General Topics, 24, 1883-1888, 1990.

Lindaas, J., Pollack, I. B., Calahorrano, J. J., O'Dell, K., Garofalo, L. A., Pothier, M. A., Farmer, D. K., Kreidenweis, S. M., Campos, T., Flocke, F., Weinheimer, A. J., Montzka, D. D., Tyndall, G. S., Apel, E. C., Hills, A. J., Hornbrook, R. S., Palm, B. B., Peng, Q., Thornton, J. A., Permar, W., Wielgasz, C., Hu, L., Pierce, J. R., Collett, J. L., Jr., Sullivan, A. P., and Fischer, E. V.: Empirical Insights Into the Fate of Ammonia in Western US Wildfire Smoke Plumes, J. Geophys. Res.-Atmos., 126, 10.1029/2020jd033730, 2021.

Liu, L. J. S., Burton, R., Wilson, W. E., and Koutrakis, P.: Comparison of aerosol acidity in urban and semirural environments, Atmos. Environ., 30, 1237-1245, 10.1016/1352-2310(95)00438-6, 1996.

Liu, T. and Abbatt, J. P. D.: Oxidation of sulfur dioxide by nitrogen dioxide accelerated at the interface of deliquesced aerosol particles, Nature Chemistry, 13, 1173-+, 10.1038/s41557-021-00777-0, 2021.

Nault, B. A., Campuzano-Jost, P., Day, D. A., Jo, D. S., Schroder, J. C., Allen, H. M., Bahreini, R., Bian, H., Blake, D. R., Chin, M., Clegg, S. L., Colarco, P. R., Crounse, J. D., Cubison, M. J., DeCarlo, P. F., Dibb, J. E., Diskin, G. S., Hodzic, A., Hu, W., Katich, J. M., Kim, M. J., Kodros, J. K., Kupc, A., Lopez-Hilfiker, F. D., Marais, E. A., Middlebrook, A. M., Andrew Neuman, J., Nowak, J. B., Palm, B. B., Paulot, F., Pierce, J. R., Schill, G. P., Scheuer, E., Thornton, J. A., Tsigaridis, K., Wennberg, P. O., Williamson, C. J., and Jimenez, J. L.: Chemical transport models often underestimate inorganic aerosol acidity in remote regions of the atmosphere, Communications Earth & Environment, 2, 10.1038/s43247-021-00164-0, 2021.

Wang, G., Kawamura, K., Hatakeyama, S., Takami, A., Li, H., and Wang, W.: Aircraft measurement of organic aerosols over China, Environ. Sci. Technol., 41, 3115-3120, 10.1021/es062601h, 2007.

Wang, G., Kawamura, K., Xie, M., Hu, S., Gao, S., Cao, J., An, Z., and Wang, Z.: Size-distributions of n-alkanes, PAHs and hopanes and their sources in the urban, mountain and marine atmospheres over East Asia, Atmos. Chem. Phys., 9, 8869-8882, 10.5194/acp-9-8869-2009, 2009.

Wang, G., Li, J., Cheng, C., Hu, S., Xie, M., Gao, S., Zhou, B., Dai, W., Cao, J., and An, Z.: Observation of atmospheric aerosols at Mt. Hua and Mt. Tai in central and east China during spring 2009-Part 1: EC, OC and inorganic ions, Atmos. Chem. Phys., 11, 4221-4235, 10.5194/acp-11-4221-2011, 2011.

Wang, G., Zhang, R., Gomez, M. E., Yang, L., Zamora, M. L., Hu, M., Lin, Y., Peng, J., Guo, S., Meng, J., Li, J., Cheng, C., Hu, T., Ren, Y., Wang, Y., Gao, J., Cao, J., An, Z., Zhou, W., Li, G., Wang, J., Tian, P., Marrero-Ortiz, W., Secrest, J., Du, Z., Zheng, J., Shang, D., Zeng, L., Shao, M., Wang, W., Huang, Y., Wang, Y., Zhu, Y., Li, Y., Hu, J., Pan, B., Cai, L., Cheng, Y., Ji, Y., Zhang, F., Rosenfeld, D., Liss, P. S., Duce, R. A., Kolb, C. E., and Molina, M. J.: Persistent sulfate formation from London Fog to Chinese haze, Proc. Natl. Acad. Sci. USA, 113, 13630-13635,

10.1073/pnas.1616540113, 2016.

Wang, H., Zhu, B., Shen, L., Xu, H., An, J., Xue, G., and Cao, J.: Water-soluble ions in atmospheric aerosols measured in five sites in the Yangtze River Delta, China: Size-fractionated, seasonal variations and sources, Atmos. Environ., 123, 370-379, 10.1016/j.atmosenv.2015.05.070, 2015.

Wu, C., Wang, G., Li, J., Li, J., Cao, C., Ge, S., Xie, Y., Chen, J., Li, X., Xue, G., Wang, X., Zhao, Z., and Cao, F.: The characteristics of atmospheric brown carbon in Xi'an, inland China: sources, size distributions and optical properties, Atmos. Chem. Phys., 20, 2017-2030, 10.5194/acp-20-2017-2020, 2020.

Yang, Y., Zhou, R., Yu, Y., Yan, Y., Liu, Y., Di, Y. a., Wu, D., and Zhang, W.: Size-resolved aerosol water-soluble ions at a regional background station of Beijing, Tianjin, and Hebei, North China, J. Environ. Sci., 55, 146-156, 10.1016/j.jes.2016.07.012, 2017.

Zhang, S., Xu, X., Lei, Y., Li, D., Wang, Y., Liu, S., Wu, C., Ge, S., and Wang, G.: Smog chamber simulation on heterogeneous reaction of O3 and NO2 on black carbon under various relative humidity conditions, Sci. Total Environ., 823, 10.1016/j.scitotenv.2022.153649, 2022.

---

## Author Response (AR2)

After reading the comments from you and the reviewers, we have carefully revised our manuscript, and highlighted the changes in yellow. Our responses to the comments are itemized below.

Anything for our paper, please feel free to contact Prof. Gehui Wang via ghwang@geo.ecnu.edu.cn.

All the best

Can Wu On behalf of Prof. Gehui Wang November 11, 2022

Reviewer(s)' Comments to Author:

**Reviewer 1**

**Comments:**

I appreciate the authors' revisions to the manuscript and believe it is stronger. In particular, the adding of some meteorological evidence has enhanced the persuasiveness of different physicochemical behaviors between nitrate and ammonium during transport. I think the manuscript is nearly ready for publication and only have minor comments for the authors to further consider.

**Reply**: We thank the reviewer's valuable comments. We have carefully revised our manuscript according to your advice.

**1. Comments:**

Please unify the format, "NH3" or "ammonia".

**Reply**: Suggestion taken. See page 2, 57-58; page19-20, line 411-437; page 21, line 463; page 24, line 520, 528; page 26, 557;

**2. Comments:**

Line 270, Please change "Xian" to "Xi'an".

**Reply**: Suggestion taken. See page 12, line 270.

**3. Comments:**

Line 290-291, When describing the site, it is suggested to use "Mountain foot" and "Mountainside" uniformly.

**Reply**: Suggestion taken. We have used the unified expressions for the both sampling sites in the revised manuscript according reviewer's advice. See page 2, line 45-60; page 13, line 290, 294; page 14, line 313; page 23, line 500; page 25, line 538;

**4. Comments:**

**Line 319, What's the differences in $PM_{2.5}$ components at MS site during daylight hours and nocturnal?**

**Reply**: From Figure 7 of the manuscript, we can note that the SNA mainly existed as  $(NH_4)_2SO_4$  and  $NH_4NO_3$  both in the daytime and at night, indicating an insignificant difference in the composition of PM2.5 that we mainly concerned in this study. However, the sources of daytime and nocturnal PM2.5 were different as illustrated the CWT analysis (Figure 4). Above results have been added in the revised manuscript. See page 20, line 431-432.

**5. Comments:**

*Line 389, What's the relationship between "lower sampling resolution" and "different diurnal cycles between SNA and PM*2.5 *at the MF"*?

**Reply**: As shown in the Figure 3, the hourly PM2.5 concentrations exhibited a morning peak at MF site. While, the daily maximum of SNA that collected at 4-hr intervals occurred at 8:00-12:00 LST (Figure S6). This time lag could partially be a result of the lower sampling resolution of SNA that may mask the subtle variation trend in this period.

**6. Comments:**

Line 487 and Line 521, It's contradictory between "NH3 emitted from wildfire would be transported aloft and lead to a higher NH3 and HNO3 mixing ratio compared to that at lower elevation" and "the source of ammonia sources is unchanged between MS and MF sites".

**Reply**: The descriptions in line 481-487 are the observation results of the wildfire smoke plumes in the western U.S. (Lindaas et al., 2021), rather than our study. Here we cited this research just for comparing the  $P_{HNO3} \times P_{NH3}/Kp$  ratios. During the our campaign, we think that the sources of ammonia was unchanged in the vertical transport process, this can be further verified by organic compounds in the  $PM_{2.5}$  .As revealed by previous studies (Wang et al., 2006; Wu et al., 2020), the levoglucosan, BkF and IP+BghiP can be used as the tracer for biomass burning, coal combustion and vehicle exhausts, respectively. From Figure S2, the difference in diagnostic ratios and proportion of these organic tracers was indistinctive among two sites. This was indicative of an insignificant change of the corresponding emission sources during the transport. Thus, we think that it is not contradictory between these two descriptions.

**7. Comments:**

**Figure 7: Better to add* $r^2$ *and p values.**

**Reply**: Suggestion taken. See page 38, line 936.

**References**

Lindaas, J., Pollack, I. B., Calahorrano, J. J., O'Dell, K., Garofalo, L. A., Pothier, M. A., Farmer, D. K., Kreidenweis, S. M., Campos, T., Flocke, F., Weinheimer, A. J., Montzka, D. D., Tyndall, G. S., Apel, E. C., Hills, A. J., Hornbrook, R. S., Palm, B. B., Peng, Q., Thornton, J. A., Permar, W., Wielgasz, C., Hu, L., Pierce, J. R., Collett, J. L., Jr., Sullivan, A. P., and Fischer, E. V.: Empirical Insights Into the Fate of Ammonia in

Western US Wildfire Smoke Plumes, J. Geophys. Res.-Atmos., 126, 10.1029/2020jd033730, 2021. Wang, G., Kawamura, K., Lee, S., Ho, K., and Cao, J.: Molecular, seasonal, and spatial distributions of organic aerosols from fourteen Chinese cities, Environ. Sci. Technol., 40, 4619-4625, 10.1021/es060291x, 2006.

Wu, C., Wang, G., Li, J., Li, J., Cao, C., Ge, S., Xie, Y., Chen, J., Li, X., Xue, G., Wang, X., Zhao, Z., and Cao, F.: The characteristics of atmospheric brown carbon in Xi'an, inland China: sources, size distributions and optical properties, Atmos. Chem. Phys., 20, 2017-2030, 10.5194/acp-20-2017-2020, 2020.